# StructTest: Benchmarking LLMs' Reasoning through Compositional Structured Outputs

## Abstract

The rapid advancement of large language models (LLMs) demands robust, unbiased, and scalable evaluation methods. However, human annotations are costly to scale, model-based evaluations are susceptible to stylistic biases, and target-answer-based benchmarks are vulnerable to data contamination and cheating. We propose StructTest, a novel benchmark that evaluates LLMs on their ability to follow compositional instructions and generate structured outputs, providing an unbiased, cost-effective, and difficult-to-cheat evaluation framework. The tasks in StructTest require significant reasoning skills. Assessments are conducted deterministically using rule-based evaluators, which can be easily extended to new tasks and datasets. By testing structured outputs across diverse domains—including Summarization, Code, HTML, and Math—and evaluating 17 popular LLMs, we demonstrate that StructTest remains challenging even for top-performing models like Deepseek-V3/R1 and GPT-4o, establishing it as a robust proxy for measuring reasoning capabilities. We believe StructTest offers a critical and complementary approach to achieving objective and comprehensive model evaluation. Our code and data are available at https://anonymous.4open.science/r/StructTest-EF37

## 1 Introduction

Since the launch of ChatGPT, the development of LLMs has accelerated dramatically—not only have general-purpose models proliferated, but reasoning capabilities have also seen significant improvements. To showcase advancements, many of these models have relied on established benchmarks like MMLU (Hendrycks et al., 2020), GSM8K (Cobbe et al., 2021a), HLE (Phan et al., 2025), Mind2Web 2 (Gou et al., 2025), GPQA (Rein et al., 2023), and LiveCodeBench (Jain et al., 2025). However, these benchmarks exhibit three major limitations: (1) human annotations are expensive to obtain, maintain, and scale; (2) model-based evaluations introduce biases inherent to the scoring model itself; and (3) answer-key–based datasets are vulnerable to data contamination. Consequently, there are needs for benchmarks that can be implemented with low cost, remain free of evaluation bias, and resist data contamination.

We introduce **StructTest**, a benchmark that evaluates *compositional instruction-following* by requiring models to generate *structured outputs*. Each task within StructTest is a ⟨ Domain Task, Format Rules ⟩ pair. The Domain Task sets the core objective (e.g., *summarize a text*, *simulate code execution*), while the Format Rules impose precise structural constraints on the output (e.g., *a hierarchical summary with a fixed number of points* or *a JSON object with predefined keys*). While StructTest's primary purpose is to assess a model's capability to interpret complex, multi-step instructions and produce correctly formatted outputs, by design, succeeding at these tasks demands significant **reasoning skills**: models must decompose tasks into subtasks, maintain internal states, enforce multiple constraints, perform abstract and meta-level reasoning, and execute logical operations in sequence. Therefore, performance on StructTest not only measures faithfulness to complex instructions but also serves as a strong proxy for a model's underlying reasoning capabilities, a correlation we demonstrate in Section 5.2.

StructTest is built on several key principles that make it an efficient, robust, and scalable benchmark:

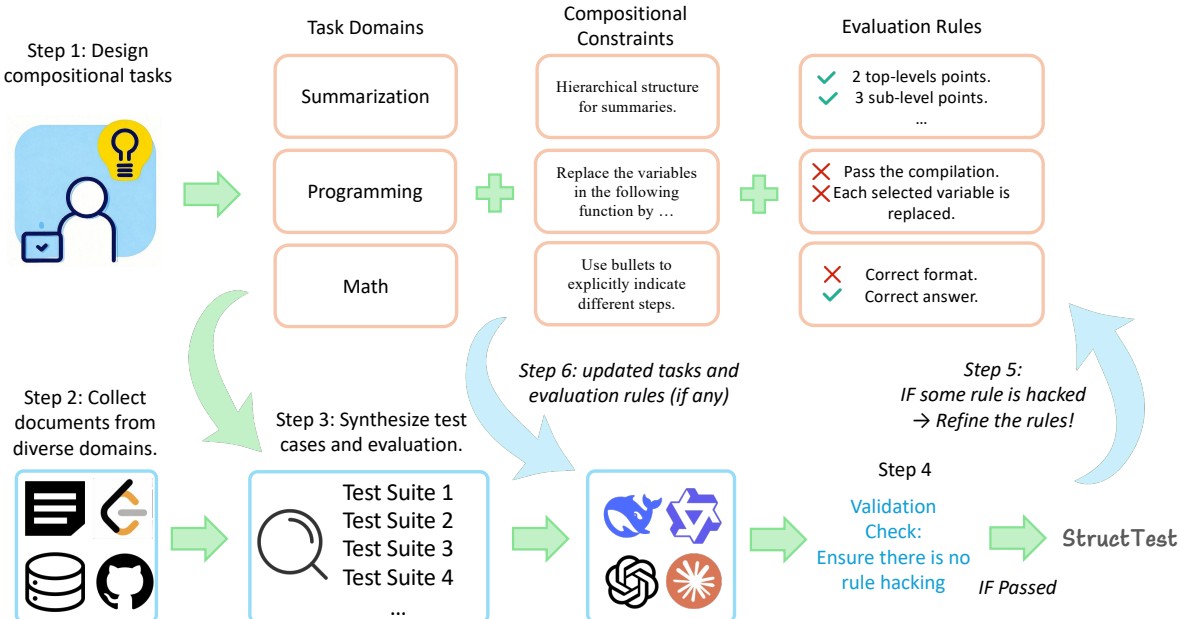

Figure 1: The construction process of StructTest is designed for adaptability, extendability, and scalability. By creating compositional tasks that are independent of underlying data and using rule-based evaluation programs, we can dynamically adjust difficulty levels and update assessments to handle new cases. This flexible, data-decoupled design ensures that StructTest remains a robust and relevant benchmark over time.

- **Reasoning Difficulty:** Tasks are inherently compositional, allowing difficulty to be scaled by increasing instruction depth. Successfully solving these tasks demands a range of reasoning skills, including constraint satisfaction, procedural reasoning, dynamic state tracking, and abstract or meta-level reasoning. This design ensures the benchmark remains a robust challenge for future generations of LLMs.

- **Programmatic Evaluation:** Assessments are fully programmatic using rules, ensuring they are deterministic, unbiased, cost-effective, and efficient.

- **Data Decoupling:** The benchmark is decoupled from underlying task data. This flexibility makes it robust to data contamination and easy to extend with new test sets and novel tasks (Section 5.1).

Figure 1 outlines the workflow for building and maintaining StructTest, which covers multiple task domains, including text summarization, code, HTML, and math. Our evaluation of 17 popular LLMs using StructTest yields several key findings: (a) **Even Top Models Struggle:** The most advanced models, including DeepSeek-v3 and R1, show significant weaknesses, especially on difficult tasks like Code-Hard and HTML-Hard

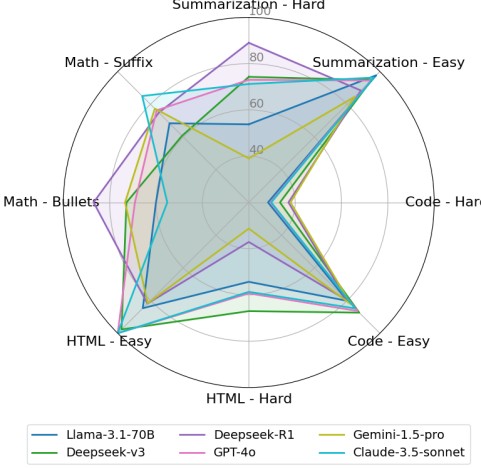

Figure 2: Comparison of top models on StructTest. DeepSeek-v3 and R1 consistently outperform others, achieving the highest scores across nearly all challenging benchmarks. However, the benchmark remains quite challenging even for these models (see Code-Hard and HTML-Hard for example).

(Figure 2). (b) **Memorization Over Generalization:** The reliance on underlying data from existing benchmarks in some subtasks, coupled with notable performance drops, raises concerns about data memorization rather than genuine reasoning generalization in current LLMs. (c) **Validated as a Reasoning Proxy:** Despite its uniqueness, StructTest is validated as a reliable proxy for general reasoning, achieving a Pearson

correlation of over 92% with both ChatBot Arena (Chiang et al., 2024) and MMLU. Its design also ensures strong extensibility and robustness against data contamination.

## 2 Literature Review

The evaluation of LLMs has emerged as a crucial research focus, especially as these models are applied to diverse tasks demanding reasoning. Current evaluation methods can be categorized into three categories: human-based, model-based, and target-answer-based. While each provides valuable insights, they also come with significant limitations.

**Human-Based Evaluation Benchmarks** A prominent example of human-based evaluation is **Chatbot Arena** (Chiang et al., 2024), which uses human voting to calculate model ELO scores. While it provides reliable assessments, it faces significant limitations: high resource costs due to extensive human annotations, limited scalability to only a few models, and challenges in sustaining community engagement for evaluating the latest models.

**Model-Based Evaluation Benchmarks** Model-based evaluation frameworks leverage LLM-as-a-judge to assess the capabilities of other models. Notable examples include **MT-Bench** (Zheng et al., 2023), **AlpacaEval** (Dubois et al., 2024), **Arena-Hard-Auto** (Li et al., 2024), and **Fofo** (Xia et al., 2024). While these frameworks offer flexibility in evaluating diverse tasks, they are prone to biases. Most notable biases include: (1) **Length Bias**: A significant issue with LLM judges is a bias towards longer responses. For example, Dubois et al. (2024) found that the GPT-4 based auto-evaluator for the popular AlpacaEval benchmark unfairly favors verbose answers. They propose a length-controlled (LC) win rate to account for this bias. (2) **Positional Bias**: It has been demonstrated that the order in which responses are presented to LLM judges can influence their decisions in pairwise judgment tasks (Wang et al., 2023). Thus it is a common practice to swap the position of the responses and measure judge consistency. (3) **Cheating by Null-Models**: As demonstrated by Zheng et al. (2024), these benchmarks can be vulnerable to exploitation. A simple "null-model" generating constant, uninformative responses can achieve a high ranking with LC win rate in AlpacaEval, raising significant concerns about the reliability of its GPT-4-based auto-evaluator. Park et al. (2024) identify four additional biases that can affect LLM judges.

**Target-Answer-Based Evaluation Benchmarks** Target-answer-based evaluations assess model capabilities by comparing directly with reference answers. Most conventional LLM benchmarks fall into this category, including **ARC** (Clark et al., 2018), **GSM8K** (Cobbe et al., 2021a), **BIG-Bench** (Zhong et al., 2024), **AGIEval** (Zhong et al., 2024) and **MMLU** (Hendrycks et al., 2020). For instance, MMLU evaluates LLMs' reasoning abilities using curated datasets from various competitive exams. While these benchmarks are unbiased, they face a significant limitation: data contamination. The extensive use of internet-sourced datasets in pre-training LLMs often overlaps with benchmark datasets, leading to inflated performance metrics and compromising the validity of evaluations (Ravaut et al., 2024a).

**StructTest in Context** To address the limitations of existing evaluation benchmarks, we introduce StructTest, which evaluates compositional instruction-following and requires LLMs to generate structured outputs. The formatted outputs are evaluated with rule-based programs. For instance, our work is related to benchmarks that evaluate instruction following with rule-based verifications. Zhou et al. (2023) introduced a benchmark with 25 easily verifiable constraints, though it features relatively shallow compositional structure and limited domain diversity. Subsequent works like FollowBench (Jiang et al., 2023) and the benchmark from Wen et al. (2024) introduced more complex constraints, but their focus remains primarily on the general text domain. Concurrent to ours, CodeIF (Yan et al., 2025) evaluates the ability of LLMs to follow instructions during code generation. It focuses on evaluating how well LLMs adhere to task-specific instructions, encompassing diverse tasks like function synthesis, debugging, refactoring, and code explanation. While several other studies have explored how format instructions influence task performance (He et al., 2024; Do et al., 2024), StructTest goes beyond simple formatting by incorporating compositional structured outputs across diverse domains. Furthermore, we demonstrate that StructTest serves as a strong and cost-effective proxy for evaluating the reasoning capabilities of LLMs.

# 3 The StructTest Benchmark

The primary goal of StructTest is to assess an LLM's ability to follow complex instructions and generate programmatically verifiable, structured outputs that are decoupled from underlying data (Figure 1). By decoupling tasks from underlying data, we minimize data contamination risks. The benchmark comprises four main task domains—summarization, coding, HTML generation, and mathematical reasoning—each designed to be adaptable, extensible, and scalable. Crucially, successful completion of these tasks requires not just instruction following but also significant underlying reasoning capabilities.

## 3.1 Summarization

As the first task in StructTest, we focus on summarization, a well-established domain for evaluating LLMs. Most existing research emphasizes the content of summaries, assessing aspects such as coherence (Chang et al., 2023), faithfulness to the source (Laban et al., 2023), coverage of diverse information (Huang et al., 2023), positional bias in context utilization (Ravaut et al., 2024b), and hallucination (Wan et al., 2024). As LLMs advance, addressing complex user requirements for summaries becomes increasingly critical. For instance, Liu et al. (2023) benchmark LLMs on content-specific instructions. However, an equally important yet underexplored aspect is the *style* or *format* of summaries. We address this gap by introducing three format-following summarization tasks. To succeed, a model must comprehend complex instructions and satisfy all enforced formatting constraints. We will first present the primary format requirements, followed by more complex and compositional versions. We provide examples for each task in Section 7.2 in the Appendix.

- **Length** Controlling summary length has been extensively researched (Liu et al., 2018; 2022). Users looking for more granular details will prompt the system to output longer summaries. To measure length-following ability, we verify whether the LLM's output $\boldsymbol{y}$ contains the required number of sentences $N$, which is sampled uniformly from a fixed interval across data points. Formally:

$$\text{Score} = \begin{cases} 1, & \text{if len}(\boldsymbol{y}) = N, \\ 0, & \text{otherwise.} \end{cases} \tag{1}$$

- **Bullet points** Bullet points are a natural method to summarize and have yielded several of the most widely used datasets in summarization research (Hermann et al., 2015; Mukherjee et al., 2022). This format is appealing to users who wish to see a clear separation of ideas in the output summary. We prompt the LLM to summarize through a list of either *unnumbered* bullet (or other symbol) points, or *numbered* points, with a varying number of points (sampled uniformly from a fixed interval).

For unnumbered points, we check whether the output contains the specified symbol $S$ in the correct number of times $N$:

$$\text{Score} = \begin{cases} 1, & \text{if count}(S \in \boldsymbol{y}) = N, \\ 0, & \text{otherwise.} \end{cases} \tag{2}$$

For numbered points, we verify that output lines $(\boldsymbol{y}_i, .., \boldsymbol{y}_M)$ are of the appropriate count and start with the correctly ordered sequence of numbers:

$$\text{Score} = \begin{cases} 1, & \text{if } (M = N) \wedge (\forall i \in [1, N], \boldsymbol{y}_{i,0} = \text{str}(i)) \\ 0, & \text{otherwise.} \end{cases} \tag{3}$$

- **Questions** Yet another approach to summarization consists in answering key questions about the source, most notably the 5 Wh-questions of what/why/who/when/where. Question-answering is a popular paradigm in summarization evaluation (Deutsch et al., 2021; Scialom et al., 2021; Fabbri et al., 2021), as it naturally enables to review that key facts from the source are covered. To induce format following, we prompt the LLM to structure its summary such that it is composed of the list of 5 Wh-questions, each followed by its corresponding answer (see Figure 10 for an example). This process is akin to query-focused summarization (Vig et al., 2022), where the Wh-questions form the query.

To evaluate Wh-questions summary formatting, we check that summary lines start with the Wh-questions. We also enforce that all questions are present, in any order. Formally, following the previous notation and noting $\boldsymbol{Q}$ the set of Wh-questions:

$$\text{Score} = \begin{cases} 1, & \text{if } (\boldsymbol{Q} \subset \boldsymbol{y}) \wedge (\forall i \in [1, N], \boldsymbol{y}_{i,0} \in \boldsymbol{Q}) \\ 0, & \text{otherwise.} \end{cases} \tag{4}$$

**Creating More Complex Tasks**

StructTest instructions following one of the aforementioned summarization formats are referred to as **Easy Summarization**. To evaluate the compositional reasoning capability of LLMs, we compose different format instructions together. We use two types of combinations: first, we specify the number of bullet points or numbered points and the desired length (in sentences) of each point; second, we ask the LLM to nest bullet points within existing points, where nested points start with the tab symbol marking indentation. These instructions combining two summarization formats are referred to as **Hard Summarization**. In this latter case, evaluation metrics defined above are also combined together and the LLM needs to verify each property.

**Data Synthesis** We collect summarization source inputs from existing trusted long-input summarization datasets. Namely, we use BigPatent (Sharma et al., 2019) and GovReport (Huang et al., 2021) from the legal domain, Arxiv and PubMed (Cohan et al., 2018) from the scientific papers domain, SummScreen (Chen et al., 2021) for screenplay summarization and QMSum (Zhong et al., 2021) for meeting summarization. In all cases, we only consider the source document and discard the ground-truth summary. For each dataset, we randomly sample 200 records from the training set, with a source length comprised between 1,500 and 15,000 words, and a ground-truth length between 75 and 750 words. Even though the ground-truth was discarded, we enforce the length criteria on it to ensure a rich enough content to be summarized.

**Remark** Unlike prior work (Liu et al., 2023), our emphasis is on format-following and generating structured output, rather than on the summary's content. This design facilitates objective, rule-based evaluation. Successfully completing these tasks requires more than just **structural and syntactic reasoning**; it demands several sophisticated capabilities: (a) **Compositional reasoning:** The model must combine multiple instructions, such as "format as a list AND create X points AND make each point Y sentences long." (b) **Multi-Constraint satisfaction:** It must hold numerous constraints (e.g., counts, lengths, formats) in its "working memory" and satisfy them all simultaneously. (c) **Procedural reasoning** It needs to follow an implicit algorithm, like the step-by-step process required for creating a nested list. (d) **Meta-reasoning:** In some cases, it requires higher-level skills like self-verification to ensure its output matches all rules.

## 3.2 Code

Given the success and widespread adoption of Code-LLMs in real-world systems (Jimenez et al., 2024; Xie et al., 2024), the ability to understand complex instructions in a language-and-code environment is critical. Programming languages offer an ideal testbed for this, as their structured nature and rule-based syntax provide a clear measure of an LLM's instruction-following capabilities. The correctness of generated code can be efficiently validated with compilers and interpreters, offering scalable, binary feedback. This makes coding a practical benchmark for structured output generation. To specifically measure task decomposition and program execution, we have developed the following tasks:

• **Add 'print' statements** We propose a code editing task where the LLM must add a print statement each time a new variable is initialized. This task is more complex than simple code completion as it requires the model to apply conditional logic across the entire input, also it probes the model's ability to maintain state—it must track which variables have already been declared in the current scope to identify only new initializations. This combination of procedural logic and state management makes it a strong test of reasoning.

Since the instruction is fixed, we can programmatically generate the expected code snippet. Specifically, we use the `ast` package[1] to parse the abstract syntax tree and extract variable initializations. The expected

---

[1] https://docs.python.org/3/library/ast.html

target code is then synthesized by inserting print statements using predefined templates. The evaluation metric is **exact match**, comparing the predicted code snippet with the synthesized one.

- **Replace variables** Another edit-based task involves *replacing variables.* For data construction, we first use the `ast` package to extract variables from a code snippet and randomly generate meaningless strings as target variable names. We then create a mapping from the original variable names to the target ones and include this mapping in the instruction, asking the LLM to replace all instances of the source variables with the corresponding target variables. The expected code snippet is generated by performing string replacements according to the mapping.

The LLM's output is evaluated using **exact match**, comparing its prediction with the synthesized expected program. This task evaluates an LLM's ability to act like a programmatic **refactoring tool**, testing whether it can go beyond generating plausible-looking code to perform precise, rule-based, and context-aware modifications.

- **Test case input generation** As a fundamental aspect of software engineering, writing high-quality unit tests (i.e., sample input-output pairs) is crucial for verifying program correctness. Given that predicting unit test outputs remains challenging for current LLMs (Li et al., 2022; Jain et al., 2024; Jiao et al., 2024), we simplify the task by asking LLMs to generate 5 distinct groups of test case inputs for a given programming problem and its corresponding solution. Successfully generating these inputs requires the model to think like a **software tester** — understand both the problem description and the provided solution and its constraints, to reason abstractly about the problem's logic (e.g., if the solution sorts a list of numbers, the LLM needs to reason that [], [3], [1, 2], [2, 1], and [0, -10] are all conceptually different and important test cases).

We evaluate the validity by executing the program on the predicted test case inputs, and if no runtime error is raised for all inputs, the generation is deemed correct. We use the **averaged pass rate** over all problems as the evaluation metric.

- **Simulate program execution** Simulating program execution requires an LLM to behave like a meticulous, rule-based machine. It must combine procedural execution with a dynamic internal "memory" that keeps track of the current value of every variable to accurately predict the program's output. This is closely tied to reasoning and agent-based operations, making program simulation a valuable proxy for assessing the ability to follow compositional instructions and perform reasoning. We prompt the LLM to simulate the step-by-step execution of a given program with specific inputs and derive the expected output. The task is divided into two difficulty levels—**Easy** and **Hard**—based on the length of the code snippet being simulated.

For the Easy level, we include multiple test cases from the original dataset for each question to ensure robust evaluation. If all predicted outputs **exactly match** the ground-truth ones, the generation for the question is considered as correct. For the Hard level, we evaluate using only one simple test case, as (1) the complexity of the code snippets themselves is sufficiently challenging, and (2) scaling test cases uniformly is difficult—some may involve millions of input numbers in a single line. The final metric is the average **exact match** rate across all questions. For all tasks, the **Easy** set contains code snippets with 3 to 30 lines, while the **Hard** set includes snippets with 50 to 200 lines.

**Data Synthesis** In order to be reliable, we collect the code snippets from existing verified code benchmarks. However, any public code snippets with paired test cases can be used for constructing the test samples. For Easy level problems, the snippets come from MBPP (Austin et al., 2021), which involves tiny segments with number of lines less than 30. For the Hard split, the samples are sourced from APPs (Hendrycks et al., 2021a) with more than 50 lines, which also comprises some much complex algorithms like recursion and dynamic programming.

**Remark** Concurrent work, CodeIF (Yan et al., 2025), evaluates how well LLMs can follow problem-specific instructions when generating code. It uses GPT-4 to both generate code instructions in the form of constraints (e.g., variable name and type) and act as an LLM-as-judge for evaluation. While comprehensive, this approach is prone to inherent biases. StructTest differs fundamentally by focusing on higher-level, reasoning-intensive tasks that are programmatically verifiable, thereby avoiding the subjectivity of LLM-based evaluation.

### 3.3 HTML Generation

The use of LLMs in generating websites has been recognized as a valuable task, reducing the workload for web designers and developers while democratizing web development for non-technical users (Calò & De Russis, 2023). In these applications, adhering to user-specified HTML structures is critical. Tang et al. (2023) highlight that LLMs often struggle to generate structured HTML, though their study is limited to simple structures and relies on content-based evaluation requiring human assessment.

In contrast, we formulate this task as to generate a specific number of standard HTML tags ("html", "head", "title", "div", "body", "h1", "h2", "p","footer") as instructed with the following structural constraints: "title" should be nested inside "head", "div" and "footer" are nested inside "body", and the rest of the tags are nested inside "div"; see Fig. 18 in the Appendix for an example. The counts of each tag to be generated are sampled uniformly from a fixed interval. Based on the range of the interval, we create two sets, **Easy** where the interval range is 2-5, and **Hard** where the range is 2-12.

We consider a generation to be successful if the count of the tags is equal to the ones provided in the prompt taking into account their nested structure and all the tags are properly formatted, i.e., an opened HTML tag has to be closed. To successfully complete this task, the LLM needs to understand the constraints and properly manage its internal states so that the generation will satisfy the structural constraints.

### 3.4 Math Reasoning

Mathematical reasoning is a common task in LLM evaluations, with benchmarks like GSM8K and MATH (Gao et al., 2024; Cobbe et al., 2021b; Hendrycks et al., 2021b). However, the influence of varying format templates on these tasks is often overlooked, potentially leading to inconsistencies, as many studies may not use impartial templates (Yu et al., 2023; Shao et al., 2024; Wei et al., 2022; Toshniwal et al., 2024). The variability in solutions—ranging from numbers and fractions to LaTeX expressions—means extraction methods may differ across studies, resulting in biased comparisons that favor models optimized for specific frameworks. For instance, MetaMathQA (Yu et al., 2023) created a dataset where answers follow specific phrases which their evaluation procedure uses to extract answers, disadvantaging models that do not adhere to these phrases.

A reliable model should not only produce correct answers but also consistently present a chain of thought or CoT (Wei et al., 2022) in a predefined format. Reliably extracting reasoning steps can be advantageous, such as for generating thought chains for process supervision (Lightman et al., 2023). Therefore, we structure our math evaluations around two key aspects: final answer parsing and CoT bullet point formatting.

• **Final answer parsing** We designed 7 distinct formats for final answer presentation and created prompts to instruct models to follow these formats. We implemented evaluation rules to assess whether a model's response adheres to the assigned format. As the source of data, we used the **GSM8K** (Cobbe et al., 2021a) benchmark, which is a set of high-school math questions. We assign each question a random format. This allows us to measure format consistency accuracy, which, when combined with math accuracy, provides a fairer and more comprehensive comparison across LLMs. Specifically, only answers that are both mathematically correct and format-compliant are considered correct. In our setup, final answer parsing is categorized as **Easy**.

• **CoT bullet points.** Solutions to math problems often involve multiple reasoning steps, and we designed 5 distinct presentation styles. These include Markdown formats like "**Step 1** ...", and JSON structures. We also defined a range for the number of steps, requiring models to adjust step granularity. For simpler solutions, models should break down steps into finer details, while for complex solutions, they should consolidate multiple steps into longer ones. By pairing each bullet point style with a unique final answer style, we created 20 formats, classified as **Hard**. We hypothesize that some LLMs may find these styles intuitive, while others may struggle, potentially leading to significant performance variations (see Section 4). Although the number of styles could theoretically be infinite, we rely on manually crafted styles to ensure accuracy and consistency.

| LLM | Average | | | Summarization | | Code | | HTML | | Math | |
|---|---|---|---|---|---|---|---|---|---|---|---|
| | All | Easy | Hard | Easy | Hard | Easy | Hard | Easy | Hard | Easy | Hard |
| DS-Dis-Qwen-1.5B | 9.19 | 14.31 | 4.07 | 26.90 | 10.08 | 24.58 | 4.21 | 0.00 | 0.00 | 5.76 | 1.97 |
| Phi-3-mini-128k | 19.97 | 32.65 | 7.30 | 55.83 | 11.39 | 51.56 | 13.33 | 0.00 | 0.00 | 23.20 | 4.47 |
| Qwen-2-7B | 17.55 | 28.58 | 6.53 | 48.79 | 9.50 | 50.63 | 13.27 | 0.33 | 0.00 | 14.56 | 3.34 |
| Mistral-7B | 13.82 | 21.83 | 5.81 | 51.29 | 15.89 | 31.25 | 5.96 | 1.67 | 0.33 | 3.11 | 1.06 |
| Llama-3.1-8B | 37.20 | 46.05 | 28.35 | 87.23 | 52.50 | 50.42 | 17.29 | 12.00 | 0.33 | 34.57 | 43.29 |
| Mistral-nemo | 27.20 | 43.76 | 10.64 | 72.83 | 18.36 | 63.13 | 17.81 | 6.33 | 0.00 | 32.75 | 6.37 |
| Mixtral-8x7B | 17.73 | 28.93 | 6.54 | 67.38 | 16.78 | 33.33 | 3.97 | 3.33 | 0.33 | 11.68 | 5.08 |
| Llama-3.1-70B | 65.93 | 82.75 | 49.10 | **97.73** | 53.75 | 80.21 | 28.29 | 84.67 | 54.33 | 68.39 | 60.05 |
| DeepSeek-v3 | 73.52 | **85.16** | 61.88 | 94.85 | 74.28 | **87.40** | 33.45 | **97.67** | **67.00** | 60.73 | 72.78 |
| DeepSeek-R1 | **74.76** | 82.42 | **67.10** | 88.42 | **89.00** | 81.85 | **37.11** | 86.67 | 55.33 | **74.91** | **86.96** |
| GPT-3.5-turbo | 38.27 | 61.75 | 14.79 | 86.35 | 22.33 | 74.48 | 19.38 | 47.67 | 6.00 | 38.51 | 11.45 |
| GPT-4o-mini | 60.04 | 75.93 | 44.16 | 98.83 | 75.58 | 82.40 | 25.67 | 45.00 | 7.67 | 77.48 | 67.70 |
| GPT-4o | **73.16** | 89.04 | **57.29** | 94.54 | 73.00 | **86.36** | 29.34 | 99.00 | 57.67 | 76.27 | 69.14 |
| Gemini-1.5-pro | 63.44 | 81.44 | 45.44 | 84.58 | 39.03 | 82.19 | **38.01** | 81.67 | 31.33 | 77.33 | **73.39** |
| Claude-3-haiku | 36.15 | 53.31 | 18.99 | 72.19 | 22.06 | 66.25 | 22.18 | 41.00 | 10.33 | 33.81 | 21.38 |
| Claude-3-opus | 68.81 | 89.14 | 48.48 | 91.21 | 46.19 | 85.00 | 36.04 | **100.00** | 56.67 | 80.36 | 55.04 |
| Claude-3.5-sonnet | 72.62 | **91.55** | 53.69 | 96.33 | 71.19 | 84.79 | 29.70 | **100.00** | **58.67** | **85.06** | 55.19 |

Table 1: Overview of results on StructTest. The best results under each task are in **bold**, and the second-best are underlined. DS-Dis-Qwen-1.5B refers to DeepSeek-R1-Distill-Qwen-1.5B. DeepSeek-R1 demonstrates strong performance, even surpassing some closed-source models on Hard. However, there remains significant room for improvement, highlighting the challenging nature of StructTest.

In addition to the **mathematical reasoning** required for a correct answer, these tasks require an LLM to employ: (a) **compositional reasoning** (combining final answer and CoT styles), (b) **constraint satisfaction** (adhering to step counts), and (c) **meta reasoning** (adjusting step granularity based on its solution path).

**Remark** Our evaluation of math-related tasks assesses both the final answer and format correctness. This dual-focus design allows us to compare a model's performance on our benchmark against its original score (e.g., on GSM8K). A performance degradation on our tasks indicates that the model struggles with format-following, revealing a "format bias" (Do et al., 2024).

## 4 Experimental Results

We evaluated StructTest against a representative selection of open-source and closed-source models; Section 7.1 in the Appendix give details about the model versions. Table 1 summarizes the overall results across all domains of StructTest for all LLMs. For open-source models, we used their instruction-tuned versions rather than the base model. Notably, the top-performing LLM, DeepSeek-R1, achieves only 74.76% accuracy on StructTest-All and 67.10% on StructTest-Hard, underscoring the benchmark's high level of difficulty. Additionally, GPT-4o is a close runner-up, and closed-source LLMs generally outperform open-source ones. In the following, we present detailed results for each domain.

• **Summarization Results** As seen in Table 1, overall, on summarization tasks, the Llama-3.1 and DeepSeek series stand out among open-source models, delivering performance comparable to GPT-4 on the Easy subset (e.g., 97.73 for Llama-3.1-70B vs. 98.83 for GPT-4o-mini). On average, closed-source LLMs outperform open-source models, especially on the Hard subset. The exception is DeepSeek-v3 and R1, which excel even beyond closed-source models on the Hard split.

Open-source LLMs other than DeepSeek experience a 70% drop in accuracy on Hard tasks compared to Easy ones, while closed-source models show a 55% relative decline. This performance gap showcases the significant challenge that LLMs face in adhering to more complex formatting instructions.

When breaking down performance across the individual summarization tasks shown in Table 2, we notice that generating numbered points is easier for LLMs than bullet points. Although all LLMs seemingly master producing numbered points, adding a constraint on the length of each point proves much harder: performance is divided by a factor of 4 for many open-source LLMs (e.g. Mistral-7B, DS-Dis-Qwen-1.5B). Indenting points

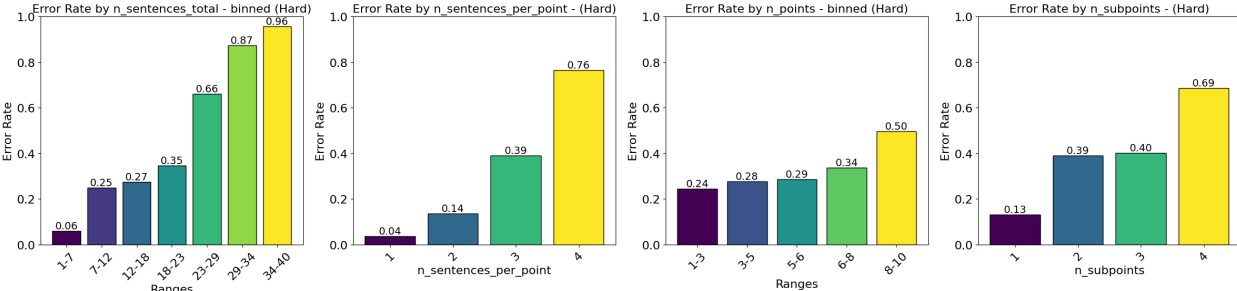

Figure 3: Error rate of GPT-4o across different features of the Summarization Bullet Points+Length (Hard) task. As specific organizational requirements are added, the error rate rises significantly.

proves to be the hardest task and half the LLMs (including closed-source GPT-3.5-turbo and Claude-3-haiku) collapse to near-zero accuracy. This shows that these models lack compositional, procedural and meta reasoning skills required to solve these tasks. DeepSeek-R1 shows a strong lead on all LLMs for all three Hard summarization tasks.

A further analysis with GPT-4o in Figure 3 shows error rate for binned values of the Hard formatting condition *Bullet points + length* (controlling the length of each bullet point). Length control error rate jumps beyond 20 total sentences, or 4 sentences per point. This finding proves that longer outputs are hard to structure and format for LLMs.

• **Code Results** As we compare the models in Table 1, we observe that DeepSeek-v3 achieves the best performance among open-source models, which can be attributed to its larger parameter size and higher-quality pretraining. DeepSeek-R1 also delivers competitive results, closely followed by Llama-3.1-70B. Among the closed-source models, Claude-3.5-sonnet and Claude-3-opus stand out as the top performers.

| LLM | Easy | | | | Hard | | |
|---|---|---|---|---|---|---|---|
| | Length | Bullet points | Numbered points | Wh-questions | Bullets + length | Numbers + length | Indented points |
| DS-Dis-Qwen-1.5B | 1.00 | 9.33 | 87.92 | 9.33 | 8.25 | 21.75 | 0.25 |
| Phi-3-mini-128k | 32.25 | 30.25 | 88.08 | 72.75 | 9.58 | 23.92 | 0.67 |
| Qwen-2-7B | 27.25 | 68.25 | 99.67 | 0.00 | 20.00 | 8.42 | 0.08 |
| Mistral-7B | 27.67 | 57.50 | 99.67 | 20.33 | 19.75 | 27.50 | 0.42 |
| Llama-3.1-8B | 93.83 | 99.67 | 99.00 | 56.42 | 62.25 | 61.75 | 33.50 |
| Mistral-nemo | 53.25 | 94.58 | 98.92 | 44.58 | 26.08 | 27.42 | 1.58 |
| Mixtral-8x7B | 33.33 | 72.50 | 86.25 | 77.42 | 19.17 | 30.25 | 0.92 |
| Llama-3.1-70B | **94.75** | **99.92** | **99.92** | 96.33 | 64.17 | 64.83 | 32.25 |
| DeepSeek-v3 | 81.33 | 98.92 | 99.17 | **100.00** | 72.58 | 73.08 | 77.17 |
| DeepSeek-R1 | 83.08 | 94.50 | 99.17 | 76.92 | **89.33** | **92.08** | **85.58** |
| GPT-3.5-turbo | 48.42 | 99.67 | **99.92** | 97.42 | 26.08 | 32.58 | 8.33 |
| GPT-4o-mini | **97.25** | 99.92 | **99.92** | 98.25 | **75.33** | **76.83** | 74.58 |
| GPT-4o | 82.92 | **100.00** | 95.25 | **100.00** | 70.25 | 76.33 | 72.42 |
| Gemini-1.5-pro | 66.50 | 99.42 | 99.50 | 72.92 | 41.00 | 23.08 | 53.00 |
| Claude-3-haiku | 67.25 | 99.33 | 99.75 | 22.42 | 29.25 | 32.08 | 4.83 |
| Claude-3-opus | 65.58 | 99.67 | 99.58 | **100.00** | 54.08 | 56.33 | 28.17 |
| Claude-3.5-sonnet | 85.58 | 99.83 | **99.92** | **100.00** | 66.50 | 66.17 | **80.92** |

Table 2: Performance comparison across LLMs on the **summarization tasks**. On the **Easy** tasks, both strong open-source and closed-source models achieve high performance. However, on the **Hard** split, DeepSeek-V3 and R1 exhibit superior instruction-following capabilities, outperforming all other models by a significant margin. Among closed-source models, GPT-4o-mini and GPT-4o show better performance than other models.

From the perspective of the individual tasks shown in Table 3, we observe that Hard-level problems are significantly more complex, as longer code snippets increase the difficulty of comprehension and procedural reasoning. Additionally, tasks requiring deeper understanding and dynamic state tracking present greater challenges. For instance, on the Easy level of *Add Print Statements* and *Replace Variables*, even smaller open-source models like Llama-3.1-8B achieve strong performance. However, only a few powerful closed-source models (e.g., the Claude series models, Gemini-1.5-pro, and GPT-4o ) perform well on the Hard level of *Replace Variables*. Furthermore, nearly all models struggle with *Test Case Inputs Generation* at the Hard level, as longer code snippets often involve multiple complex operations, such as loops, recursion, and switch-case statements. On the Hard split *Simulate Execution*, we find most smaller models fail to pass more than 20% questions. Among the close-sourced models, Gemini-1.5-pro demonstrates significantly better performance but lags behind DeepSeek-R1.

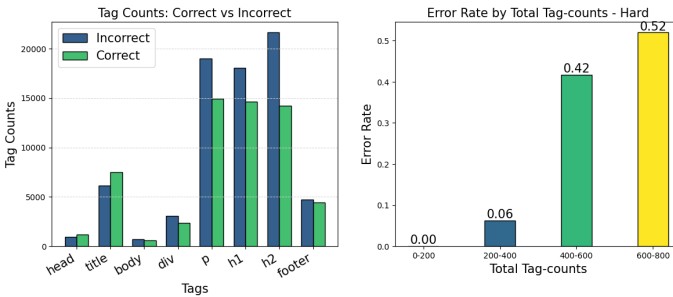

Figure 4: Tag-counts for correct vs. incorrect HTML generations (left) and error rate by total tag counts (binned) (right) for the Hard task in GPT-4o.

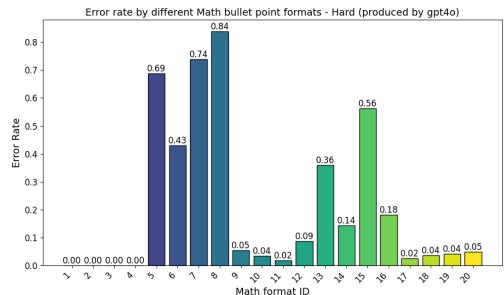

Figure 5: Error rates of GPT-4o in GSM8K math reasoning across 20 Hard formats.

• **HTML Results** From the results in Table 1, we observe that open-source models generally underperform compared to closed-source ones in both Easy and Hard HTML generation tasks, with higher accuracies in the Easy task than in the Hard task. Among open-source models, DeepSeek-v3 leads, followed by DeepSeek-R1, while among closed-source models, Claude-3.5-sonnet is the top performer, closely trailed by Claude-3-opus and GPT-4o. DeepSeek-v3, DeepSeek-R1, Claude-3.5-sonnet, Claude-3-opus, and GPT-4o also rank among the highest in MMLU scores (Table 5). Additionally, larger models generally outperform smaller ones, as seen with Llama-3.1-70B compared to Llama-3.1-8B.

| LLM | Add Print | | Replace Vars | | Input Gen | | Simulate Exec | |
|---|---|---|---|---|---|---|---|---|
| | Easy | Hard | Easy | Hard | Easy | Hard | Easy | Hard |
| DS-Dis-Qwen-1.5B | 32.50 | 0.0 | 32.08 | 7.29 | 6.67 | 0.00 | 49.93 | 9.55 |
| Phi-3-mini-128k | 70.42 | 0.50 | 84.58 | 43.28 | 12.50 | 0.00 | 38.75 | 9.55 |
| Qwen-2-7B | 60.83 | 1.01 | 79.17 | 41.00 | 22.92 | 0.00 | 39.58 | 11.05 |
| Mistral-7B | 47.08 | 0.50 | 25.00 | 17.31 | 33.75 | 0.00 | 19.17 | 6.03 |
| Llama-3.1-8B | 77.50 | 2.01 | 85.83 | 53.08 | 1.66 | 3.52 | 36.67 | 10.55 |
| Mistral-nemo | 74.17 | 2.51 | 82.08 | 52.16 | 50.42 | 0.00 | 45.83 | 16.58 |
| Mixtral-8x7B | 40.42 | 0.50 | 12.50 | 9.34 | 40.83 | 0.50 | 39.58 | 5.53 |
| Llama-3.1-70B | 95.00 | 21.61 | 87.92 | 64.92 | 66.67 | 4.02 | 71.25 | 22.61 |
| DeepSeek-v3 | **96.25** | **24.12** | 88.33 | **69.48** | **73.75** | 5.53 | 91.25 | 34.67 |
| DeepSeek-R1 | 95.00 | 22.11 | 89.58 | 66.52 | 50.00 | 2.01 | 84.17 | **57.79** |
| GPT-3.5-turbo | 76.25 | 0.00 | 90.42 | 57.40 | 72.92 | 1.51 | 58.33 | 18.59 |
| GPT-4o-mini | 90.00 | 10.55 | 91.25 | 66.51 | 66.25 | 3.02 | 82.08 | 22.61 |
| GPT-4o | 85.00 | 9.55 | 86.67 | 70.62 | **79.58** | 4.52 | 94.17 | 32.66 |
| Gemini-1.5-pro | 94.17 | 34.17 | 83.33 | 70.62 | 65.83 | 4.02 | 85.42 | **43.22** |
| Claude-3-haiku | 75.42 | 5.03 | 86.67 | 60.59 | 40.00 | 5.53 | 62.92 | 17.59 |
| Claude-3-opus | 96.25 | 40.20 | 91.67 | 78.82 | 69.58 | 2.01 | 82.50 | 23.12 |
| Claude-3.5-sonnet | 90.00 | 9.55 | 91.25 | 78.59 | 70.42 | **6.03** | 87.50 | 24.62 |

Table 3: Performance comparison across LLMs on **code-related tasks** shows that among open-source models, DeepSeek-v3 and R1 excel, particularly on the hard split. Among closed models, Gemini-1.5-pro, Claude-3-opus, and Claude-3.5-sonnet achieve higher accuracy.

We further provide two analyses based on GPT-4o's performance on the Hard task in Figure 4: one examines the distribution of cumulative tag-counts for each tag in both correct and incorrect HTML code generation samples, and the other analyzes the distribution of all tag-counts in incorrect HTML code generation samples. Both figures reveal a consistent trend of increasing error rates as the number of tag-counts grows, confirming that LLMs struggle with structured HTML code generation, particularly when tasked with producing a larger number of HTML tags. This issue is especially pronounced for deeply nested tags like "div", "p", "h1", and "h2", as these tags are generated multiple times more frequently than their parent containers due to their nesting structure. This shows that even the best LLMs struggle with multiple constraints and state tracking.

• **Math Results** In Table 1 we present the math format-following percentage accuracy for the Easy (final answer style) and Hard (final answer and bullet point style) categories, using GSM8K as the underlying benchmark. To be considered correct, an answer must be both accurate and compliant with the corresponding format requirement. As mentioned before, By evaluating both semantic and format correctness, we can directly compare a model's performance on our benchmark against its original score.

Most models perform significantly worse in both settings compared to their scores in standard benchmarks (Gao et al., 2024). For instance, Gemini-1.5-pro achieves 77.33% in Easy and 73.39% in Hard, while scoring 91.7% in the original dataset. In fact, while most closed-source models in Table 1 exceed 90% in standard benchmarks (Gao et al., 2024), they experience significant performance drops in our evaluations, with margins

| LLM | Add Print | | Replace Vars | | Input Gen | | Simulate Exec | |
|---|---|---|---|---|---|---|---|---|
| | Obfuscation | Normal | Obfuscated | Normal | Obfuscated | Normal | Obfuscated | Normal |
| DS-Dis-Qwen-1.5B | 40.00 | 32.50 | 26.67 | 32.08 | 5.83 | 6.67 | 16.67 | 49.93 |
| Phi-3-mini-128k | 44.58 | 70.42 | 62.08 | 84.58 | 1.67 | 12.50 | 35.00 | 38.75 |
| Qwen-2-7B | 62.92 | 60.83 | 75.00 | 79.17 | 16.25 | 22.92 | 32.92 | 39.58 |
| Mistral-7B | 47.08 | 47.08 | 31.67 | 25.00 | 33.75 | 33.75 | 14.17 | 19.17 |
| Llama-3.1-8B | 74.17 | 77.50 | 80.42 | 85.83 | 15.84 | 1.66 | 33.33 | 36.67 |
| Mistral-nemo | 72.08 | 74.17 | 80.83 | 82.08 | 43.75 | 50.42 | 42.92 | 45.83 |
| Mixtral-8x7B | 53.75 | 40.42 | 7.50 | 12.50 | 34.58 | 40.83 | 34.17 | 39.58 |
| Llama-3.1-70B | 95.00 | 95.00 | 85.42 | 87.92 | 65.42 | 66.67 | 69.58 | 71.25 |
| DeepSeek-V3 | **97.08** | **96.25** | **93.33** | 88.33 | **70.00** | **73.75** | 84.17 | **91.25** |
| DeepSeek-R1 | 95.42 | 95.00 | 91.25 | **89.58** | 57.50 | 50.00 | **91.25** | 84.17 |

Table 4: Comparison between normal data with obfuscated data under the easy split of code task. Few models exhibit significant performance degradation, e.g., Phi-3-mini-128k-instruct, deepseek-r1-1.5b, and the relative rankings between different models remains stable.

as high as 70%. This indicates that these models are not as reliably or consistently proficient in math-related formats and may have overfitted to specific formats and styles. Notably, smaller and older closed-source models like GPT-3.5-turbo and Claude-3-haiku show considerable degradation, with scores below 40%. Most open-source models, such as Mixtral-8x7B, perform even worse, dropping below 10% accuracy on the Hard split. This indicates these models lack constraint satisfaction and meta reasoning skills required for these Hard tasks. However, DeepSeek-R1 demonstrates stronger resilience in maintaining format compliance, likely due to its deep reasoning process. Overall, these results suggest that existing math reasoning comparisons between models are likely unreliable and unfair unless tested across a wide variety of diverse and impartial formats. Our framework offers a more robust alternative for such evaluations.

To provide deeper insights, Figure 5 illustrates the error rates of GPT-4o on GSM8K when tested across 20 Hard formats. Despite being a highly advanced frontier model, GPT-4o exhibits widely varying performance. Specifically, it achieves perfect scores (zero error rate) in formats 1 to 4 but struggles in others, with error rates as high as 84%. This suggests that the model may have overfitted to certain popular formats while faltering with novel ones. We conducted manual inspection and found that the model actually often produces accurate final answers but fails to adhere to the instructed formats, resulting in these samples being marked as incorrect. In those cases, the format in which the model committed to, however, is inconsistent and unpredictable, leading to parsing difficulty to conduct further checks.

## 5  Discussion and Further Analysis

### 5.1  Robustness to Contamination and Benchmark Scalability

A key challenge in benchmarking LLMs is data contamination, where models inadvertently train on test data. StructTest mitigates this risk in two primary ways.

First, the design of StructTest inherently minimizes data contamination. By focusing on novel, compositional tasks that demand higher-level reasoning (e.g., meta reasoning and state tracking), we create problems unlikely to exist in training data. We verified this by creating an **obfuscated version** of our coding tasks where we replaced variables with random strings and changed function names counterfactually (e.g., if a prompt requests a function for quick sort, we would rename the function signature from `quick_sort` to `get_max`) to eliminate any potential shortcuts. The results are shown in Table 4. The fact that most models showed minimal performance degradation on this obfuscated data confirms that the original StructTest is already robust against contamination and effectively evaluates reasoning rather than memorization.

Second, the benchmark is designed as a living, adaptable evaluation. Its flexible framework allows for the periodic introduction of new tasks, domains, and complexity levels. To future-proof StructTest, we will maintain a confidential, held-out test set that is regularly updated. This approach safeguards the benchmark

| LLM | StructTest | Arena | MMLU |
|---|---|---|---|
| Phi-3-mini-128k | 18.81 | 1,037 | 68.10 |
| Mistral-7B | 14.08 | 1,072 | 60.10 |
| Llama-3.1-8B | 37.22 | 1,175 | 73.00 |
| Mixtral-8x7B | 18.12 | 1,114 | 70.60 |
| Llama-3.1-70B | 65.69 | 1,248 | 86.00 |
| DeepSeek-V3 | 73.66 | 1,319 | 88.50 |
| DeepSeek-R1 | 72.15 | 1,361 | 90.80 |
| GPT-3.5-turbo | 38.27 | 1,068 | 70.00 |
| GPT-4o-mini | 60.04 | 1,272 | 82.00 |
| GPT-4o | 73.50 | 1,265 | 88.70 |
| Gemini-1.5-pro | 63.44 | 1,302 | 85.90 |
| Claude-3-haiku | 36.15 | 1,179 | 75.20 |
| Claude-3-opus | 68.81 | 1,247 | 86.80 |
| Claude-3.5-sonnet | 72.62 | 1,268 | 88.70 |

Table 5: Comparison of StructTest average accuracy with ChatBot Arena scores and MMLU accuracy. ChatBot Arena results are current as of March 13th, 2025.

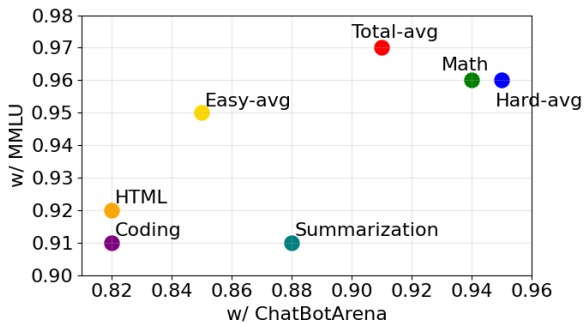

Figure 6: Correlation of various StructTest setups with ChatBot Arena and MMLU. The Hard and Total splits show strong correlation with both MMLU and ChatBot Arena, with math demonstrating the highest correlation among the four domains.

against potential rule-hacking, ensures that performance reflects true generalization capabilities, and provides a scalable, cost-efficient method for evaluating new models over time.

## 5.2 Correlation to General Reasoning

To determine if StructTest can serve as a cost-effective yet strong proxy for general reasoning in LLMs, we compared its average accuracy with scores from two popular benchmarks: ChatBot Arena and MMLU. The results are shown in Table 5. We include all the models for which we could find both Arena and MMLU scores. The correlation (Pearson's product-moment coefficient) between StructTest and Arena is **92.5**%, while the correlation with MMLU is **96.3**%. These results highlight that StructTest—despite being unbiased, inexpensive to evaluate, and robust to data contamination—yields results that are strongly correlated with resource-intensive benchmarks.

A more detailed analysis, presented in the 2D scatter plot in Figure 6, breaks down this correlation by task domain and difficulty. The results show that the Total and Hard splits maintain a high correlation with both MMLU and Chatbot Arena, while the Math domain has the highest correlation. This suggests that StructTest can effectively serve as a reliable proxy for general reasoning, with the added benefits of being easily extendable and free from the risks of data contamination and evaluation bias.

## 5.3 Discussion about Updating StructTest On-the-fly

As LLMs rapidly advance, benchmarks must be dynamically updated to ensure their reliability and mitigate data contamination. StructTest was designed from its inception for this kind of evolution, though its long-term success will depend on collaborative community efforts. We envision updates across three key areas:

**Rules** Despite extensive efforts to scrutinize model predictions, some cases may still arise where models exploit loopholes to artificially inflate scores. When such cases are identified, the corresponding evaluation rules can be updated to enhance robustness.

**Tasks** Similar to rule updates, new tasks can be introduced by carefully designing evaluation rules. This aspect relies more on community contributions. When failures are identified in powerful models, these patterns can be analyzed to develop new tasks for StructTest, ensuring its continued relevance and challenge.

**Data** Most tasks in StructTest do not rely on annotated benchmark data, and they are decoupled from underlying data by rule-based evaluation. This enables the seamless integration of newly collected raw corpora, further minimizing the risk of data contamination.

# 6 Conclusion

We have proposed StructTest, a programmatically verifiable benchmark for evaluating instruction-following capabilities of LLMs through structured outputs. StructTest is a cheap-to-run and unbiased benchmark with adjustable difficulty levels, which is especially robust to the prevailing issue of contamination among existing LLM benchmarks. By conducting evaluation across 17 popular LLMs, we find that it remains challenging even for the very best models like DeepSeek-R1, GPT-4o or Claude-3.5-sonnet, which all score below 70% accuracy on the Hard subset of StructTest. Notably, the open-source DeepSeek-R1 shines in StructTest as its performance is comparable to the top closed-source models. Besides, lower results on the math domain compared with those on the standardized benchmarks reveal the potential overfitting to answer format of existing LLMs. Our analysis of correlation with other benchmarks (i.e., MMLU and ChatBot Arena) shows that StructTest serves as a good proxy for evaluating general reasoning ability in LLMs. We believe that StructTest offers a critical, complementary approach to existing LLM evaluations.

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

# 7 Appendix

## 7.1 Closed-Source Model Versions

We show the API version used in our evaluation results for close-source models in Section 7.1. The inference for all closed-source models was performed during 27th November 2024 to 14th December 2024.

| Model | API Version |
|---|---|
| GPT-3.5-turbo | gpt-3.5-turbo-0125 |
| GPT-4o-mini | gpt-4o-mini-2024-07-18 |
| GPT-4o | gpt-4o-2024-08-06 |
| Gemini-1.5-pro | gemini-1.5-pro-002 |
| Claude-3-haiku | claude-3-haiku-20240307 |
| Claude-3-opus | claude-3-opus-20240229 |
| Claude-3.5-sonnet | claude-3-5-sonnet-20241022 |

Table 6: Closed-source model versions used in Evaluation Results

## 7.2 Examples for Different Tasks

We show examples for each summarization task in Figures 7 to 13, and examples for each coding task in Figures 14 to 17.

```
## Instruction

The following is a source document.

Source:
one surprising result that has come out of the more than 200 extrasolar planet discoveries to date is the wide range
of eccentricities observed . unlike our own solar system , many of the extrasolar planets which are not tidally
locked to their host stars have moderate eccentricities ( @xmath1 ) , and 15 planets have high eccentricities (
@xmath0 ) . these observations have spawned several theories as to the origin of highly eccentric extrasolar planets
. one such method , planet - planet scattering , occurs when multiple jovian planets form several astronomical units
( au ) from the host star and then interact , leaving one in an eccentric orbit and often ejecting the other @xcite .
(truncated)

Format Instruction: Please summarize the above text in 5 sentences.{Place the summary between <summary>
and </summary>.

Summary:

#### Response

<summary>
The discovery of over 200 extrasolar planets has revealed a wide range of eccentricities, unlike our
solar system, with many planets exhibiting moderate to high eccentricities. Various theories have been
proposed to explain these eccentricities, including planet-planet scattering and interactions with
stellar companions. This study focuses on six highly eccentric planetary systems (HD 3651, HD 37605, HD
45350, HD 80606, HD 89744, and 16 Cyg B) using observational data and dynamical simulations to explore
the possibility of additional planets. The results show no evidence for additional planets in these
systems, but the simulations indicate regions where additional planets could exist in stable orbits. The
study also sets detection limits for potential additional planets, ruling out short-period planets down
to a few Neptune masses in some systems.
</summary>
```

Figure 7: Test example for **length** task in Summarization.

```
## Instruction

The following is a source document.

Source:

supersymmetry ( susy ) is one of the most attractive extensions of the standard model . this symmetry
solves the naturalness problem and predicts gauge coupling unification at the gut scale @xmath1 . it
also predicts the existence of superpartner of the standard model ( sm ) particles . from the
naturalness argument , their masses should be below tev range , hence these particles will be discovered
at tevatron or large hadron collider ( lhc ) . (truncated)

Format Instruction: Please summarize the above text in {n_points} bullet points using the following
symbol: * to start each bullet point.
Place the summary between <summary> and </summary>. Please don't generate any introductory comments.
Summary:

#### Response

<summary>
* Supersymmetry (SUSY) is an extension of the Standard Model that addresses the naturalness problem,
predicts gauge coupling unification at the GUT scale, and suggests the existence of superpartners for
Standard Model particles, which should be detectable at the Tevatron or LHC.
* The no-scale type Kähler potential, where the hidden and observable sectors are separated, results in
all scalar masses and trilinear scalar couplings vanishing as the cosmological constant vanishes, making
gaugino masses the only source of SUSY breaking.
* Three scenarios for SUSY particle mass spectrum are considered: universal gaugino masses at the GUT
scale, universal gaugino masses above the GUT scale, and non-universal gaugino masses at the GUT scale.
The minimal scenario is severely constrained by cosmological and direct search bounds, while the other
two scenarios offer more viable parameter regions.
* Current constraints from the lightest Higgs mass and the branching ratio for \( b \rightarrow s \gamma
\) further limit the parameter space, with the minimal scenario being almost entirely excluded. The
non-universal gaugino mass scenario allows for wino-like or higgsino-like neutralino as the LSP, which
remains cosmologically viable.
</summary>
```

Figure 8: Test example for **bullet points** task in Summarization.

```
## Instruction

The following is a source document.

Source:

x - ray studies of fairly normal " galaxies , with high - energy emission not obviously dominated by a luminous active galactic
nucleus ( agn ) , have recently been extended to cosmologically interesting distances in the deep field ( cdf ) surveys , which
have now reached 1 ms of exposure ( cdf - n : hornschemeier et al . 2001 , hereafter paper ii ; brandt et al . 2001b , hereafter
paper v ; cdf - s : tozzi et al . 2001 ; p. rosati et al . , in prep . ) . galaxies with @xmath8 are detected in appreciable
numbers at 0.52 kev fluxes below @xmath9 erg @xmath6 s@xmath7 ( e.g. , paper ii ) ; the cdf - n survey goes almost two orders of
magnitude fainter , detecting significant numbers of normal galaxies among the population of x - ray sources making the diffuse x
- ray background ( xrb ; paper ii ; a.j . barger et al . , in prep . ) . (truncated)

Format Instruction: Please summarize the above text in 4 numbered points where each point starts with a number and numbers follow
the correct increasing order.
Place the summary between <summary> and </summary>. Please don't generate any introductory comments.

Summary:

#### Response

<summary>
1. Recent deep field (CDF) surveys have extended X-ray studies of normal galaxies to cosmologically significant distances,
detecting significant numbers of these galaxies at faint flux levels and contributing to the diffuse X-ray background (XRB).

2. The study focuses on the evolution of the X-ray luminosity to optical luminosity ratio of normal spiral galaxies up to redshift
z ≈ 1.5, using spectroscopic and photometric redshift samples to filter galaxies by optical luminosity and morphology.

3. Stacking analysis of X-ray data from the CDF-N survey reveals significant detections in the soft X-ray band for galaxies up to
z ≈ 1.5, with the average X-ray luminosity of these galaxies showing a slight increase with redshift, consistent with heightened
star formation rates in the past.

4. The study finds that normal spiral galaxies contribute up to 5-10% of the XRB flux in the 0.5-2 keV band, with their average
X-ray luminosities evolving by a factor of up to 2.5 by z ≈ 1.5, providing constraints on models of X-ray binary production and
star formation history in galaxies.
</summary>
```

Figure 9: Test example for **numbered points** task in Summarization.

```
## Instruction

The following is a source document.

Source:

one surprising result that has come out of the more than 200 extrasolar planet discoveries to date is the wide range of
eccentricities observed . unlike our own solar system , many of the extrasolar planets which are not tidally locked to their host
stars have moderate eccentricities ( @xmath1 ) , and 15 planets have high eccentricities ( @xmath0 ) . these observations have
spawned several theories as to the origin of highly eccentric extrasolar planets . one such method , planet - planet scattering ,
occurs when multiple jovian planets form several astronomical units ( au ) from the host star and then interact , leaving one in
an eccentric orbit and often ejecting the other @xcite . (truncated)

Place the summary between <summary> and </summary>. Please don't generate any introductory comments.

Summary:

#### Response

<summary>
[What is the main point?]: The main point is the investigation of the presence of additional low-mass planets in six highly
eccentric extrasolar planetary systems using observational data and dynamical simulations.

[Why is it happening?]: This research is happening to understand the architecture of planetary systems, particularly those with
highly eccentric orbits, and to determine where additional planets could exist in stable orbits.

[Who is involved?]: The study involves astronomers and researchers using data from the Hobby-Eberly Telescope (HET) and other
observatories, with contributions from various institutions including the University of Texas at Austin, Pennsylvania State
University, and others.

[When is the action happening?]: The observations and data collection span several years, with specific data points mentioned from
2004 to 2007.

[Where is the action happening?]: The action is happening at multiple observatories, primarily the Hobby-Eberly Telescope (HET) at
McDonald Observatory, and also includes data from the Harlan J. Smith Telescope and other sources.
</summary>
```

Figure 10: Test example for **questions** task in Summarization.

```
## Instruction

The following is a source document.

Source:

the transport properties of nonlinear non - equilibrium dynamical systems are far from well - understood@xcite . \n consider in
particular so - called ratchet systems which are asymmetric periodic potentials where an ensemble of particles experience directed
transport@xcite . \n the origins of the interest in this lie in considerations about extracting useful work from unbiased noisy
fluctuations as seems to happen in biological systems@xcite . (truncated)

Format Instruction: Please summarize the above text in 8 bullet points using the following symbol: • to start each bullet point.
The total length should be 8 sentences, and each bullet point should have exactly 1 sentences. Place the summary between <summary>
and </summary>.

Summary:

#### Response

<summary>
• The transport properties of nonlinear non-equilibrium dynamical systems, particularly chaotic deterministic ratchets, are
complex and sensitive to system parameters and initial conditions.
• Mateos proposed that current reversals in such systems are linked to bifurcations from chaotic to periodic behavior, but Barbi
and Salerno challenged this, suggesting current reversals can occur without such bifurcations.
• The study investigates the impact of different particle ensembles on the current, showing that the current depends significantly
on the initial ensemble and the need to discard transients.
• The results indicate that while bifurcations often correspond to abrupt changes in the current, not all bifurcations lead to
current reversals, and not all current reversals are associated with bifurcations.
• The study finds that single trajectory results can differ significantly from ensemble averages, especially in chaotic regimes,
highlighting the importance of considering ensemble characteristics.
• The research supports a generalized version of Mateos's conjecture, suggesting that bifurcations are associated with sudden
current changes, which may or may not lead to current reversals.
• The study emphasizes the need for detailed analysis of specific systems to discover heuristic rules for understanding chaotic
ratchets.
• The findings suggest that while broad heuristics are rare, understanding ensemble dependence is critical for studying the
transport properties of chaotic systems.
</summary>
```

Figure 11: Test example for **combination of bullet points and length** task in Summarization.

```
## Instruction

The following is a source document.

Source:

the transport properties of nonlinear non - equilibrium dynamical systems are far from well - understood@xcite . \n consider in
particular so - called ratchet systems which are asymmetric periodic potentials where an ensemble of particles experience directed
transport@xcite . \n the origins of the interest in this lie in considerations about extracting useful work from unbiased noisy
fluctuations as seems to happen in biological systems@xcite . (truncated)

Format Instruction: Please summarize the above text in 8 numbered points where each point starts with a number and numbers follow
the correct increasing order. The total length should be 8 sentences, and each numbered point should have exactly 1 sentences.
Place the summary between <summary> and </summary>.

Summary:

#### Response

<summary>
<summary>
1. The transport properties of nonlinear non-equilibrium dynamical systems, particularly chaotic deterministic ratchets, are
complex and sensitive to system parameters and initial conditions.
2. Mateos proposed that current reversals in such systems are linked to bifurcations from chaotic to periodic behavior, but Barbi
and Salerno challenged this, suggesting current reversals can occur without such bifurcations.
3. The study investigates the impact of different particle ensembles on the current, showing that the current depends
significantly on the initial ensemble and the need to discard transients.
4. The results indicate that while bifurcations often correspond to abrupt changes in the current, not all bifurcations lead to
current reversals, and not all current reversals are associated with bifurcations.
5. The study finds that single trajectory results can differ significantly from ensemble averages, especially in chaotic regimes,
highlighting the importance of considering ensemble characteristics.
6. The research supports a generalized version of Mateos's conjecture, suggesting that bifurcations are associated with sudden
current changes, which may or may not lead to current reversals.
7. The study emphasizes the need for detailed analysis of specific systems to discover heuristic rules for understanding chaotic
ratchets.
8. The findings suggest that while broad heuristics are rare, understanding ensemble dependence is critical for studying the
transport properties of chaotic systems.
</summary>
```

Figure 12: Test example for **combination of numbered points and length** task in Summarization.

```
## Instruction

The following is a source document.

Source:

the transport properties of nonlinear non - equilibrium dynamical systems are far from well - understood@xcite . \n consider in
particular so - called ratchet systems which are asymmetric periodic potentials where an ensemble of particles experience directed
transport@xcite . \n the origins of the interest in this lie in considerations about extracting useful work from unbiased noisy
fluctuations as seems to happen in biological systems@xcite . (truncated)

Format Instruction: Please summarize the following text using 8 bullet points and 2 levels of indentation. Each bullet point starts
with the symbol •, repeated only once. Then, each bullet point should be followed by 1 sub-points, each starting with a tab followed
by the same symbol •, only repeated once as well. Place the summary between <summary> and </summary>.

Summary:

#### Response

<summary>
<summary>
•• The transport properties of nonlinear non-equilibrium dynamical systems, particularly ratchet systems, are complex and not
well-understood.
        •• Ratchet systems involve asymmetric periodic potentials where particles experience directed transport, with interest stemming
from extracting work from noisy fluctuations.
•• Recent studies have focused on deterministic chaotic and Hamiltonian ratchets, with chaotic systems being sensitive to initial
conditions and system parameters.
        •• The phase-space structure of these systems is complicated, with stability islands, chaotic seas, and co-existing attractors
that change with parameters.
•• Mateos proposed a model showing current direction reversal correlated with bifurcations from chaotic to periodic behavior, but this
was contested by Barbi and Salerno.
        •• Barbi and Salerno argued that current reversals could occur without such bifurcations, suggesting stability properties of
rotating periodic orbits as the cause.
•• The current study investigates the impact of different particle ensembles on the current in a chaotic deterministic ratchet system.
        •• It shows that the current depends significantly on the initial ensemble details and emphasizes the importance of discarding
transients.
•• The study finds that while bifurcations often correspond to abrupt changes in the current, not all bifurcations lead to current
reversals.
        •• It suggests that most current reversals are associated with bifurcations, but not all current changes are reversals.
•• The study highlights the importance of ensemble characteristics in understanding transport properties, especially near bifurcation
points.
        •• It proposes a heuristic that bifurcations correspond to sudden current changes, which may or may not be reversals.
•• The study concludes that while general rules of thumb are useful, the behavior of chaotic ratchets is complex and requires detailed
consideration of ensemble characteristics.
        •• Future research should aim to concretize the reasons for when these heuristics are valid or violated.
•• Acknowledgments are given to various individuals and institutions for their support and discussions.
        •• The study references several works in the field of nonlinear physics and complex systems.
</summary>
```

Figure 13: Test example for **indented bullet points** task in Summarization.

```
## Instruction

I will show you a code snippet. Please add a print statement whenever a new variable is defined,
following the format below:

print("X: ", str(x), "Y: ": str(Y), …)

where `X` and `Y` should be the newly defined variables.

### Notes:

- If there is no newly defined variables, do not add any print statements.
- If some variables that were initialized previously are assigned with new values, add print statements
for these variables after the newest assignments, too.
- Do not change any content of the other code
- Overlook the temperature variables like those defined in for loops.

### Response Format

Please include your answer within <ans> and </ans> tags.

### Example

Here is an example for your reference:

#### The code to add print statements

```
def get_last_checkpoint(folder):
    content = os.listdir(folder)
    checkpoints = [
        path
        for path in content
        if _re_checkpoint.search(path) is not None and os.path.isdir(os.path.join(folder, path))
    ]
    if len(checkpoints) == 0:
        return None
    return os.path.join(folder, max(checkpoints, key=lambda x:
int(_re_checkpoint.search(x).groups()[0])))
```

#### Response

<ans>
def get_last_checkpoint(folder):
    content = os.listdir(folder)
    print("content: ", str(content))
    checkpoints = [
        path
        for path in content
        if _re_checkpoint.search(path) is not None and os.path.isdir(os.path.join(folder, path))
    ]
    print("checkpoints: ", str(checkpoints))
    if len(checkpoints) == 0:
        return None
    return os.path.join(folder, max(checkpoints, key=lambda x:
int(_re_checkpoint.search(x).groups()[0])))
</ans>

Now, let's get started:

#### The code to add print statements

```
def remove_Occ(s,ch):
    for i in range(len(s)):
        if (s[i] == ch):
            s = s[0 : i] + s[i + 1:]
            break
    for i in range(len(s) - 1,-1,-1):
        if (s[i] == ch):
            s = s[0 : i] + s[i + 1:]
            break
    return s
```

#### Response
```

26
```

Figure 14: Test example for **add 'print' statements** task with one-shot prompting.

```
## Instruction

I will show you a code snippet. Your task is to replace the name of variables to different ones
according to the mapping I give to you, and return me back the new code snippet after replacement.

### Response format

Please inclcude your answer within <ans> and </ans> tags.

Here is an example for your reference:

#### Code Snippet

```
def get_last_checkpoint(folder):
    content = os.listdir(folder)
    checkpoints = [
        path
        for path in content
        if _re_checkpoint.search(path) is not None and os.path.isdir(os.path.join(folder, path))
    ]
    if len(checkpoints) == 0:
        return None
    return os.path.join(folder, max(checkpoints, key=lambda x:
int(_re_checkpoint.search(x).groups()[0])))
```

#### Variable Renaming

```
path -> ppp
content -> ccc
```

#### Response

<ans>
def get_last_checkpoint(folder):
    ccc = os.listdir(folder)
    checkpoints = [
        ppp
        for ppp in ccc
        if _re_checkpoint.search(ppp) is not None and os.path.isdir(os.path.join(folder, ppp))
    ]
    if len(checkpoints) == 0:
        return None
    return os.path.join(folder, max(checkpoints, key=lambda x:
int(_re_checkpoint.search(x).groups()[0])))
</ans>

Now, let's get started:

#### Code Snippet

```
def remove_Occ(s,ch):
    for i in range(len(s)):
        if (s[i] == ch):
            s = s[0 : i] + s[i + 1:]
            break
    for i in range(len(s) - 1,-1,-1):
        if (s[i] == ch):
            s = s[0 : i] + s[i + 1:]
            break
    return s
```

#### Variable Renaming

```
s -> str_var
ch -> char_var
i -> index_var
```

#### Response
```

27
```

Figure 15: Test example for **replace variables** task with one-shot prompting.

```
## Instruction

You are an expert programmer. I will show you a programming problem as well as one solution program.
Please help me to generate **5** groups of test case inputs to this function.

### Response format

1. Your test case inputs should be in the correct python object format so that we can initialize them
into an argument list by calling `func(*eval(inputs))`.
2. Separate each group of test case inputs simply by new lines.
3. Include all the generated test case inputs within `<ans>` and `</ans>` tags.

Here is an example for your reference:

#### Problem description

Your music player contains N different songs and she wants to listen to L (not necessarily different)
songs during your trip.  You create a playlist so that:

Every song is played at least once
A song can only be played again only if K other songs have been played

Return the number of possible playlists.  As the answer can be very large, return it modulo 10^9 + 7.

#### Solution program

```
def numMusicPlaylists(N: int, L: int, K: int) -> int:
    s=0
    c=0
    r=0
    x=math.factorial(N)
    while(True):
        c=x*((N-r-K)**(L-K))*(-1)**(r)//(math.factorial(N-r-K)*math.factorial(r))
        if(c!=0):
            s=(s+c)%(10**9+7)
            r+=1
        else:
            return s
```

#### Response

<ans>
[3, 3, 1]
[2, 3, 0]
[2, 3, 1]
[4, 3, 1]
[4, 2, 2]
</ans>

Now, let's get started:

#### Program description

Write a python function to remove first and last occurrence of a given character from the string.

#### Solution program

```
def remove_Occ(s,ch):
    for i in range(len(s)):
        if (s[i] == ch):
            s = s[0 : i] + s[i + 1:]
            break
    for i in range(len(s) - 1,-1,-1):
        if (s[i] == ch):
            s = s[0 : i] + s[i + 1:]
            break
    return s
```

#### Response
```

Figure 16: Test example for **test case input generation (easy)** task with one-shot prompting. Easy level

```
## Instruction

I will show you a program as well as a group of inputs. Please simulate the execution process of this function, and return me
back to the outputs.

## Response Format

Please include your final results following the expected output format within <ans> and </ans> tags.

## Notes:

- You can simulate the program step by step via arbitrary formats. Just remember to tag the final results last.
- Please follow the expected output format defined by the program.
- The output(s) should be in proper python object so that we can use `eval(x)` to initialize it/them.
- The values of different arguments are separated by commas.

Here is an example for your reference:

### Code snippet

```
def maxScore(cardPoints: List[int], k: int) -> int:
    max_score = 0
    curr_score= 0
    init_hand = cardPoints[len(cardPoints)-k:]
    max_score = sum(init_hand)
    curr_score = max_score
    for i in range(k):
        curr_score -= init_hand[i]
        curr_score += cardPoints[i]
        if curr_score > max_score:
            max_score = curr_score
    return max_score
```

### Inputs

```
[1,2,3,4,5,6,1], 3
```

### Response

Let's think step by step.
1. Initialize `max_score` and `curr_score` to 0.
2. Initialize `init_hand` to the last `k` elements of `cardPoints`. As a result, we have `init_hand = [5,6,1]`.
3. Initialize `max_score` to the sum of `init_hand`. As a result, we have `max_score = 12`.
4. Initialize `curr_score` to `max_score`. As a result, we have `curr_score = 12`.
5. Iterate over the range of `k`:
    - In the first iteration, we have `i = 0`.
    - Update `curr_score` by subtracting the first element of `init_hand` and adding the first element of `cardPoints`. As a
result, we have `curr_score = 12 - 5 + 1 = 8`.
    - Check if `curr_score` is greater than `max_score`. Since `8` is not greater than `12`, we don't update `max_score`.
    - In the second iteration, we have `i = 1`.
    - Update `curr_score` by subtracting the second element of `init_hand` and adding the second element of `cardPoints`. As a
result, we have `curr_score = 8 - 6 + 2 = 4`.
    - Check if `curr_score` is greater than `max_score`. Since `4` is not greater than `12`, we don't update `max_score`.
    - In the third iteration, we have `i = 2`.
    - Update `curr_score` by subtracting the third element of `init_hand` and adding the third element of `cardPoints`. As a
result, we have `curr_score = 4 - 1 + 3 = 6`.
    - Check if `curr_score` is greater than `max_score`. Since `6` is not greater than `12`, we don't update `max_score`.
6. Return `max_score`. As a result, we have `12`.

So the final result is
<ans>
12
</ans>
----

Now, let's get started:

### Code snippet

```
def remove_Occ(s,ch):
    for i in range(len(s)):
        if (s[i] == ch):
            s = s[0 : i] + s[i + 1:]
            break
    for i in range(len(s) - 1,-1,-1):
        if (s[i] == ch):
            s = s[0 : i] + s[i + 1:]
            break
    return s
```

### Inputs

```
"hello", "l"
```

### Response
```

```
### Instruction

[Example Instruction] Generate only an html code that has 1 html tag.
Inside the html tag generate 2 head tags and 2 body tags. Inside of
each head tag, generate 2 title  tags and inside of each body tag,
generate 2 div tags and 2 footer tags. Inside of each div tag, generate
1 h1 tag and 1 p tag. Generated code:

<html>
 <head>
   <title></title>
   <title></title>
 </head>
 <head>
   <title></title>
   <title></title>
 </head>
 <body>
   <div>
      <h1></h1>
      <p></p>
   </div>
   <footer></footer>
 </body>
 <body>
   <div>
      <h1></h1>
      <p></p>
   </div>
   <footer></footer>
 </body>
</html>

### Format Instruction

[Real Instruction]
Generate only an html code that has 1 html tag. Inside the html tag, generate
1 head tag and 1 body tag. Inside of each head tag, generate 1 title tag and inside
of each body tag, generate 2 div tags and 1 footer tag. Inside of each div tag, generate
1 h1 tag, 1 h2 tag and 1 p tag.

### Task Instruction

Please generate the full html code according to the instruction. Do not use any abbreviation.
The generated html should only contain tags, do not put any content like title
1.1, paragraph 2.1, etc. Your generated html code:

### Response

<html>
    <head>
        <title></title>
    </head>
    <body>
        <div>
            <h1></h1>
            <h2></h2>
            <p></p>
        </div>
        <div>
            <h1></h1>
            <h2></h2>
            <p></p>
        </div>
        <footer></footer>
    </body>
</html>
```

Figure 18: Test example for **html generation** task with one-shot prompting.