# OpenReview forum: "StructTest: Benchmarking LLMs’ Reasoning through Compositional Structured Outputs"
_TMLR — Withdrawn by Authors_

### Review · Reviewer_dQNZ · 2025-09-23

**Summary Of Contributions:**

This work introduces a new benchmark for testing structured output and some level of compositional reasoning in LLMs. The benchmark includes different domains: summarisation, HTML, code, and math. Moreover, it is structured in a way that makes it easy to increase the complexity of the problems.

Overall the paper is well written, although I have some remarks on clarity (see below). I believe it tackles important points: an adjustable benchmark that can be easily adapted to new developments in the field, and also to possible domain contamination. Several state-of-the-art LLMs have been tested, leading to a good pool of results that can be used to establish the benchmark usefulness.

However, I have to main problems with the paper.
1. Considering that the main point of the paper is introducing the benchmark, there is very little / unclear explanations about the details on StructTest design.
2. The conclusion that this benchmark is a good proxy for general reasoning seems a bit farfetched.

More in details:
- Figure 1 is central to the idea, and it is never described. The caption just repeats what’s in the main body already, and in the main body it’s never described how the entire pipeline works. The vague idea is clear, but I believe it should be clear how this benchmark makes it easy to extend it to new constraints / problem sizes. The code is publicly available, so one could check that directly, but the papers makes it an important point that it has been designed carefully, and this is not reflected in the text.
- In general, the text can benefit of explanatory examples of the different problems and challenges. Already on page two, “scaled by increasing instruction depth” is very vague. An example here could clarify what type of “depth” we are talking about. Also second point could be better clarified with an example what “programmatic rules” are being used. Even though in this case it’s easier to imagine.
- Later in the paper, when all the different tasks are described, a lot of details are hidden. A lot of space is devoted to the “summarisation” task, and a lot less to the rest (I would argue the equations 1-4 are not really providing an insight and could be pushed to the appendix, in some cases they make the intuition more complex than what it is in practice, I’m not sure what’s the benefit of having them). Some aspects are not clear and just vaguely introduced (e.g. the format templates in the math task, but not only). I understand there are some space issues, but understanding the results without at least an idea on how these benchmarks work, it’s quite difficult. For example, without understanding the type of formatting required in the math task, it’s hard to follow the explanation and intuitions of the corresponding results. I believe that some examples within the main body would solve the issue, while keeping the length under control.
- In section 3.2, “For the Hard level, we evaluate using only one simple test case”. It is not clear to me what this means. Can you clarify it?
- I understand that with StructTest you can adapt the benchmarks in the future, in case you notice future LLMs start exploiting them. In multiple times in the paper it is mentioned that StructTest is designed to ensure “robustness against data contamination”. However, for example in 3.4 it’s mentioned that as source of data, GSM8K is used. How is this not causing problems with the “robustness against data contamination” claim?
- About Section 5.2, I do appreciate the correlation analysis, however it is not clear to me how one can make the step “high correlation -> StructTest can measure general reasoning”.
    - First, to the best of my knowledge, Arena and MMLU are not tailored to measure general reasoning capabilities. Arena might indirectly measure it, but it’s mostly about user preference. MMLU measures it a bit more directly, but being multiple-choice is not really about open-ended / multi-step problem solving skills. It is more about recalling factual / common-sense knowledge. So, why using these two benchmarks?
    - Second, even if they measure general reasoning. Correlation does not mean that StructTest is good to measure the same capabilities. I suppose this is why it is phrased as a “proxy to..”? Indeed, correlation means that it is predictive of performance on the other benchmarks, but this correlation might be due to other variables (e.g. model size, training technique / contamination, evaluation format, …). Can you please elaborate on this?

Minor things:
- Figure 2. I assume it reports accuracy, but would be better to specify it explicitly.
- Table 1. In the caption I would mention that the horizontal dashed line splits open and closed source models. It’s clear only later / by reading the text.
- Table 4. The “obfuscated” term is explained only later in the text. I would add it to the caption such that the table is self explained.
- Table 5. Is the accuracy of StructTest an average across all the tasks? If yes, I would write it. If not, needs to be clarified.
- “LogicBench: Towards Systematic Evaluation of Logical Reasoning Ability of Large Language Models” (Parmar et al, 2024) seems like it’s relevant work
- Typo: in the remark paragraph on page 5, the closed quotes should be ‘’ in latex, not “
- What is the relationship of your work with works like “Adaptable Logical Control for Large Language Models” (Zhang et al, 2024)? Of course they are proposing a model, and not a benchmark, but it seems the type of constraints they look into is similar to the structural/syntactical constraint of StructTest. Might provide extra motivation / future work?

**Audience:**

Yes

**Audience Explanation:**

I do agree with the authors that this benchmarks is complementary to other benchmarks and offers to test capabilities of LLMs that are otherwise not tested.

**Claims And Evidence:**

No

**Claims Explanation:**

See concerns about section 5.2 in my review.

**Requested Changes:**

Authors should tackle all my concerns and answer my questions. An answer to the questions in the "minor things" part is also appreciated.

---

### Review · Reviewer_Qnuz · 2025-09-30

**Summary Of Contributions:**

This paper presents StructTest, an LLM benchmark that uses compositional instructions and structured outputs to dynamically create novel tasks of controllable difficulty.  The primary motivation for StructTest is to address 3 common challenges in today's state of the art benchmarking: expense of human annotations, bias of model-based evaluations, and leakage/contamination concerns when using static benchmarks.

Strengths:
- Well-motivated approach to building a benchmark. Tying reasoning behaviors to easily testable structured outputs is a nice
- Good empirical analyses of many large language models performance on StructTest; and additional experiments on correlations w/existing hard benchmarks, and (implied) robustness to contamination.

Weaknesses:
- Little to no discussion of tasks that are not amenable to this approach.  Is there a potential to expand compositionality methods to cover more open-ended tasks (creative writing, textual analyses, etc)
- Some StructTest rules seem amenable to reward hacking.  This is briefly acknowledged in 5.3, but not discussed in detail.
- Related to reward hacking, at the moment, StructTest results are correlated with MMLU, ChatbotArena, but it's not clear that is fundamental.  If LLMs are trained specifically for StructTest, this correlation could break. Is the functionality specifically tested by StructTest useful on its own, rather than as a proxy for more general reasoning?

Also:
- Not a weakness per se, but I hope authors can add results on current latest models (gpt-5, o4, ...)

Questions:
- "For better or for worse, benchmarks shape a field" - David Patterson.  If StructTest is successful, it will provide a target to drive improvements in LLMs. I'd be curious to hear the authors' perspective on how attempts to achieve higher scores on StructTest is likely to shape the capabilities of LLMs, and perhaps better justify these specific 3 task domains (or point at promising / important additional task domains to add to StructTest).
- How should we interpret the difference in model ranking by StructTest vs ChatbotArena / MMLU in Table 5?
- The test of robustness to contamination is suggestive, but still leaves room for models to memorize or narrowly learn specific subtasks (e.g., the model could learn to add "print()" statements but not any others). StructTest's compositionality seems to enable a strategy to address such issues. It would be great to expand/formalize section 5.3 to state a rigorous plan.

**Audience:**

Yes

**Audience Explanation:**

There is a broad interest in benchmarking LLMs and the compositional approach taken by StructTest presents benefits that would interest people.

**Broader Impact Concerns:**

None.

**Claims And Evidence:**

Yes

**Claims Explanation:**

Yes, the authors include empirical evaluation of the efficacy of StructTest benchmark in differentiating behaviors of foundation models.  Additional tests show correlation between StructTest and other related benchmarks and also experiments suggestive of a robustness to contamination.

**Requested Changes:**

- Discuss the limits of StructTest w/respect to tasks that may not be amenable to structured outputs
- In Sec 5, deepen discussion around reward hacking and contamination.  Formalize a plan for structtest updates that counteracts narrow optimization
- better justification of StructTest's targeted capabilities and task domains.  Why are these useful capabilities on their own? If an LLM gets better at StructTest but not at ChatbotArena / MMLU, is that still a benefit?
- update with latest model results, as feasible

---

### Review · Reviewer_79fc · 2025-10-10

**Summary Of Contributions:**

**Summary:** The paper proposes StructTest, a deterministic, rule-verified benchmark for compositional instruction following with structured outputs across four domains (Summarization, Code, HTML, Math). Each item pairs task content with explicit format constraints; success requires both semantic correctness and format compliance, scored programmatically with validators. The authors evaluate a broad set of open- and closed-weight LLMs, show substantial headroom, especially on hard splits, and report correlations with external leaderboards (Chatbot Arena, MMLU). They also include an obfuscation study for code to probe contamination and claim stable rankings under identifier renaming.

**Strengths:** The motivation is solid. Frontier LLMs have saturated many existing benchmarks, so an unseen and fully deterministic benchmark is genuinely needed. Evaluating the suite on top tier models is a plus, and the results show it still challenges very capable systems, which supports its practical value. The cross-domain coverage (summarization, code, HTML, math) also brings out format and state-tracking failure modes that a correctness-only setup would miss.

**Weaknesses:** The solution does not really answer the question it sets up. Most of the tasks reduce to format or style compliance rather than genuine reasoning, and this is especially true for the summarization and math tracks. The math setting is basically a format reward wrapped around familiar GSM8K-style content, not a test of mathematical cognition. These kinds of constraints are known to be easy to game, and in practice even small models can learn them quickly with a bit of RL on verifiable rewards, which makes the headline difficulty less convincing. The central claim that this benchmark can act as a proxy for true reasoning is not supported beyond simple correlations. Correlation is not causation, and you can imagine a 1B model that learns to satisfy the style rules while still lagging badly on real reasoning. The paper also blurs format compliance and semantic correctness, so it is hard to tell whether failures reflect reasoning limits or just formatting brittleness.

**Audience:**

No

**Audience Explanation:**

I do not think this will interest most TMLR readers. A deterministic suite for structured instruction following is useful, but the tasks mostly test style or format that is easy to game and does not measure reasoning. The paper does not provide head-to-head comparisons with existing benchmarks on overlapping subsets, so claims are not substantiated. The related-work justification stays qualitative and never quantifies composition depth, validator coverage, or difficulty controls, so the contribution feels incremental.

**Claims And Evidence:**

No

**Claims Explanation:**

Some parts are supported. The authors ran many strong models and used deterministic scoring, which I appreciate. But the main claim is not supported: the metrics are mostly style and format checks that are easy to game, and recent work shows 7-8B models can quickly learn to satisfy verifiable constraints with short RL or verifier-guided training, without clear gains in actual reasoning [1]. Combined with evidence of reward over-optimization in RLHF, the correlation to Arena/MMLU is not persuasive as a reasoning proxy.

[1] Pyatkin, Valentina, et al. "Generalizing Verifiable Instruction Following." arXiv preprint arXiv:2507.02833 (2025).

**Requested Changes:**

- Test whether format/style can be trained around. Run a targeted RL-from-verifiable-rewards (RLVR) intervention on StructTest tasks (summarization-format, HTML, code transforms, math-format), then re-evaluate the exact same models before vs. after. In parallel, report movement on independent reasoning targets (e.g., AIME’25-style math and LiveCodeBench) to see if StructTest gains truly track reasoning instead of format compliance. If small RLVR runs yield large StructTest jumps with little change on AIME/LiveCodeBench, that undercuts the “proxy for reasoning” claim; if they move together, it strengthens it.


- Please enumerate exact dataset sizes per task/split, not only sources. For summarization you state “randomly sample 200” per dataset, which is clear; do the same for code, HTML, and math so totals are unambiguous (train/test counts, Easy/Hard counts). Also provide full inference settings for every model: max new tokens, temperature, top-p, decoding strategy, prompt templates, and any retries. The appendix currently lists API/model versions and evaluation dates for closed-source models but not these decode parameters, which blocks faithful replication.

- Several StructTest failure modes (especially in code and HTML) plausibly vanish when models can call tools (executors, validators, linters, simple HTML checkers). Please add results for at least one open model with local tool-calling (e.g., gpt-oss 20B with python task calls) and quantify how much performance lifts when tool-use is enabled vs. disabled. This matters because many real deployments now rely on function/tool calls; a benchmark that penalizes “no-tool” agents may not reflect deployed reasoning.

- The related-work section is helpful but largely narrative. Please add side-by-side numbers on overlapping capabilities against representative instruction-following/format-verified suites (e.g., IF-Eval/Zhou et al., FollowBench, Wen et al., CodeIF for code) and explain, quantitatively, where StructTest is stricter/more diagnostic (false-positive rate to “style hacks,” compositionality depth, rule coverage).

---

### Note · Authors · 2025-10-18

**Comment:**

Dear Editor and Reviewers,

Thank you for your thoughtful reviews and constructive suggestions. After careful consideration, we have decided to withdraw this submission to substantially revise the paper and strengthen the experiments. We are grateful for your time and expertise.

Best regards,
Authors of StructTest

**Withdrawal Confirmation:**

I have read and agree with the venue's withdrawal policy on behalf of myself and my co-authors.